# MULTIMARGINAL GENERATIVE MODELING WITH STOCHASTIC INTERPOLANTS

**Michael S. Albergo**
Center for Cosmology and Particle Physics
New York University
New York, NY 10003, USA
`albergo@nyu.edu`

**Nicholas M. Boffi**
Courant Institute of Mathematical Sciences
New York University
New York, NY 10012, USA
`boffi@cims.nyu.edu`

**Michael Lindsey**
Department of Mathematics
University of California, Berkeley
Berkeley, CA 94720, USA
`lindsey@math.berkeley.edu`

**Eric Vanden-Eijnden**
Courant Institute of Mathematical Sciences
New York University
New York, NY 10012, USA
`eve2@cims.nyu.edu`

## ABSTRACT

Given a set of $K$ probability densities, we consider the multimarginal generative modeling problem of learning a joint distribution that recovers these densities as marginals. The structure of this joint distribution should identify multi-way correspondences among the prescribed marginals. We formalize an approach to this task within a generalization of the stochastic interpolant framework, leading to efficient learning algorithms built upon dynamical transport of measure. Our generative models are defined by velocity and score fields that can be characterized as the minimizers of simple quadratic objectives, and they are defined on a simplex that generalizes the time variable in the usual dynamical transport framework. The resulting transport on the simplex is influenced by all marginals, and we show that multi-way correspondences can be extracted. The identification of such correspondences has applications to style transfer, algorithmic fairness, and data decorruption. In addition, the multimarginal perspective enables an efficient algorithm for reducing the dynamical transport cost in the ordinary two-marginal setting. We demonstrate these capacities with several numerical examples.

## 1 INTRODUCTION

Generative models built upon dynamical transport of measure, in which two probability densities are connected by a learnable transformation, underlie many recent advances in unsupervised learning (Rombach et al., 2022; Dhariwal & Nichol, 2021). Contemporary methods such as normalizing flows (Rezende & Mohamed, 2015) and diffusions (Song et al., 2021b) transform samples from one density $\rho_0$ into samples from another density $\rho_1$ through an ordinary or stochastic differential equation (ODE/SDE). In such frameworks one must learn the velocity field defining the ODE/SDE. One effective algorithm for learning the velocity is based on the construction of a *stochastic interpolant*, a stochastic process that interpolates between the two probability densities at the level of the individual samples. The velocity can be characterized conveniently as the solution of a tractable square loss regression problem. In conventional generative modeling, one chooses $\rho_0$ to be an analytically tractable reference density, such as the standard normal density, while $\rho_1$ is some target density of interest, accessible only through a dataset of samples. In this setting, a general stochastic interpolant $x_t$ can be written as

$$x_t = \alpha_0(t)x_0 + \alpha_1(t)x_1, \tag{1}$$

where $x_0 \sim \rho_0$ and $x_1 \sim \rho_1$ and we allow for the possibility of dependence between $x_0$ and $x_1$. Meanwhile $\alpha_0(t)$ and $\alpha_1(t)$ are differentiable functions of $t \in [0,1]$ such that $\alpha_0(0) = \alpha_1(1) = 1$ and $\alpha_0(1) = \alpha_1(0) = 0$. These constraints guarantee that $x_{t=0} = x_0$ and $x_{t=1} = x_1$ by construction. It was shown in Albergo & Vanden-Eijnden (2023); Albergo et al. (2023) that if $X_t$ is the solution of

the ordinary differential equation (ODE)

$$\dot{X}_t = b(t, X_t), \tag{2}$$

with velocity $b$ defined by

$$b(t, x) = \mathbb{E}[\dot{x}_t | x_t = x] = \dot{\alpha}_0(t) \, \mathbb{E}[x_0 | x_t = x] + \dot{\alpha}_1(t) \, \mathbb{E}[x_1 | x_t = x] \tag{3}$$

and initial condition $X_0$ drawn from $\rho_0$, then $X_t$ matches $x_t$ in law for all times $t \in [0, 1]$. Hence we can sample from $\rho_1$ by generating samples from $\rho_0$ and propagating them via the ODE (2) to time $t = 1$. Equivalent diffusion processes depending on the score can also be obtained by the introduction of noise into the interpolant (Albergo et al., 2023).

A significant body of research on both flows and diffusions has studied how to $\alpha_0(t)$ and $\alpha_1(t)$ that reduce the computational difficulty of integrating the resulting velocity field $b$ (Karras et al., 2022).

In this work, we first observe that the decomposition of the velocity field of (1) into conditional expectations of samples from the marginals $\rho_0$ and $\rho_1$ suggests a more general definition of a process $x(\alpha)$, defined not with respect to the scalar $t \in [0, 1]$ but with respect to an *interpolation coordinate* $\alpha = (\alpha_0, \alpha_1)$,

$$x(\alpha) = \alpha_0 x_0 + \alpha_1 x_1. \tag{4}$$

By specifying a curve $\alpha(t)$, we can recover the interpolant in (1) with the identification $x_t = x(\alpha(t))$. We use the generalized perspective in (4) to identify the minimal conditions upon $\alpha$ so that the density $\rho(\alpha, x)$ of $x(\alpha)$ is well-defined for all $\alpha$, reduces to $\rho_0(x)$ for $\alpha = (1, 0)$, and to $\rho_1(x)$ for $\alpha = (0, 1)$. These considerations lead to several new advances, which we summarize as our **main contributions**:

1. We show that the introduction of an interpolation coordinate decouples the *learning problem* of estimating a given $b(t, x)$ from the *design problem* for a path $\alpha(t)$ governing the transport. We use this to devise an optimization problem over curves $\alpha(t)$ with the Benamou-Brenier transport cost, which gives a geometric algorithm for selecting a performant $\alpha$.

2. By lifting $\alpha$ to a higher-dimensional space, we use the stochastic interpolant framework to devise a generic paradigm for multimarginal generative modeling. To this end, we derive a generalized probability flow valid among $K + 1$ marginal densities $\rho_0, \rho_1, \ldots, \rho_K$, whose corresponding velocity field $b(t, x)$ is defined via the conditional expectations $\mathbb{E}[x_k | x(\alpha) = x]$ for $k = 0, 1, \ldots, K$ and a curve $\alpha(t)$. We characterize these conditional expectations as the minimizers of square loss regression problems, and show that one gives access to the score of the multimarginal density.

3. We show that the multimarginal framework allows us to solve $\binom{K}{2}$ marginal transport problems using only $K$ marginal vector fields. Moreover, this framework naturally learns multi-way correspondences between the individual marginals, as detailed in Section 3. The method makes possible concepts like all-to-all image-to-image translation, where we observe an emergent style transfer amongst the images.

4. Moreover, in contrast to existing work, we consider multimarginal transport from a novel *generative* perspective, demonstrating how to generate joint samples $(x_0, \ldots, x_K)$ matching prescribed marginal densities and generalizing beyond the training data.

The structure of the paper is organized as follows. In Section 1.1, we review some related works in multimarginal optimal transport and two-marginal generative modeling. In Section 2, we introduce the interpolation coordinate framework and detail its ramifications for multimarginal generative modeling. In Section 2.1, we formulate the path-length minimization problem and illustrate its application. In Appendix B, we consider a limiting case where the framework directly give one-step maps between any two marginals. In Section 3, we detail experimental results in all-to-all image translation and style transfer. We conclude and discuss future directions in Section 4.

## 1.1 RELATED WORKS

**Generative models and dynamical transport**   Recent years have seen an explosion of progress in generative models built upon dynamical transport of measure. These models have roots as early as (Tabak & Vanden-Eijnden, 2010; Tabak & Turner, 2013), and originally took form as a discrete series of steps (Rezende & Mohamed, 2015; Dinh et al., 2017; Huang et al., 2016; Durkan et al., 2019), while modern models are typically formulated via a continuous-time transformation. A

particularly notable example of this type is score-based diffusion (Song et al., 2021c;a; Song & Ermon, 2019), along with related methods such as denoising diffusion probabilistic models (Ho et al., 2020; Kingma et al., 2021), which generate samples by reversing a stochastic process that maps the data into samples from a Gaussian base density. Methods such as flow matching (Lipman et al., 2022; Tong et al., 2023), rectified flow (Liu, 2022; Liu et al., 2022), and stochastic interpolants (Albergo & Vanden-Eijnden, 2023; Albergo et al., 2023) refine this idea by connecting the target distribution to an arbitrary base density, rather than requiring a Gaussian base by construction of the path, and allow for paths that are more general than the one used in score-based diffusion. Importantly, methods built upon continuous-time transformations that posit a connection between densities often lead to more efficient learning problems than alternatives. Here, we continue with this formulation, but focus on generalizing the standard framework to a multimarginal setting.

**Multimarginal modeling and optimal transport**    Multimarginal problems are typically studied from an optimal transport perspective (Pass, 2014b), where practitioners are often interested in computation of the Wasserstein barycenter (Cuturi & Doucet, 2014; Agueh & Carlier, 2011; Altschuler & Boix-Adsera). The barycenter is thought to lead to useful generation of combined features from a set of datasets (Rabin et al., 2012), but its computation is expensive, leading to the need for approximate algorithms. Some recent work has begun to fuse the barycenter problem with dynamical transport, and has developd algorithms for its computation based on the diffusion Schrödinger bridge (Noble et al., 2023).

A significant body of work in multimarginal optimal transport has concerned the existence of an optimal transport plan of Monge type (Gangbo & Świech, 1998; Pass, 2014a), as well as generalized notions (Friesecke & Vögler, 2018). A Monge transport plan is a joint distribution of the restricted form (31) considered below and in particular yields a set of compatible deterministic correspondences between marginal spaces. We refer to such a set of compatible correspondences as a multi-way correspondence. Although the existence of Monge solutions in multimarginal optimal transport is open in general (Pass, 2014a), in the setting of the quadratic cost that arises in the study of Wasserstein barycenters, in fact a Monge solution does exist under mild regularity conditions (Gangbo & Świech, 1998).

We do not compute multimarginal optimal transport solutions in this work, but nonetheless we will show how to extract an intuitive multi-way correspondence from our learned transport. We will also show that if the multimarginal stochastic interpolant is defined using a coupling of Monge type, then this multi-way correspondence is uniquely determined.

## 2    MULTIMARGINAL STOCHASTIC INTERPOLANTS

In this section, we study stochastic interpolants built upon an interpolation coordinate, as illustrated in (4). We consider the multimarginal generative modeling problem, whereby we have access to a dataset of $n$ samples $\{x_k^i\}_{k=0,\ldots,K}^{i=1,\ldots,n}$ from each of $K+1$ densities with $x_k^i \sim \rho_k$. We wish to construct a generative model that enables us to push samples from any $\rho_j$ onto samples from any other $\rho_k$. By setting $\rho_0$ to be a tractable base density such as a Gaussian, we may use this model to draw samples from any marginal; for this reason, we hereafter assume $\rho_0$ to be a standard normal. We denote by $\Delta^K$ the $K$-simplex in $\mathbb{R}^{K+1}$, i.e., $\Delta^K = \{\alpha = (\alpha_0, \ldots, \alpha_K) \in \mathbb{R}^{K+1} : \sum_{k=0}^K \alpha_k = 1 \text{ and } \alpha_k \geq 0 \text{ for } k = 0, \ldots K\}$. We begin with a useful definition of a stochastic interpolant that places $\alpha \in \Delta^K$:

**Definition 1** (Barycentric stochastic interpolant). *Given $K + 1$ probability density functions $\{\rho_k\}_{k=0}^K$ with full support on $\mathbb{R}^d$, the barycentric stochastic interpolant $x(\alpha)$ with $\alpha = (\alpha_0, \ldots, \alpha_K) \in \Delta^K$ is the stochastic process defined as*

$$x(\alpha) = \sum_{k=0}^K x_k \alpha_k, \tag{5}$$

*where $(x_1, \ldots, x_K)$ are jointly drawn from a probability density $\rho(x_1, \cdots, x_K)$ such that*

$$\forall k = 1, \ldots, K \; : \; \int_{\mathbb{R}^{(K-1)\times d}} \rho(x_1, \ldots, x_K) dx_1 \cdots dx_{k-1} dx_{k+1} \cdots dx_K = \rho_k(x_k), \tag{6}$$

*and we set $x_0 \sim N(0, Id_d)$ independent of $(x_1, \ldots, x_K)$.*

The barycentric stochastic interpolant emerges as a natural extension of the two-marginal interpolant (1) with the choice $\alpha_0(t) = 1 - t$ and $\alpha_1(t) = t$. In this work, we primarily study the barycentric interpolant for convenience, but we note that our only requirement in the following discussion is that $\sum_{k=0}^{K} \alpha_k^2 > 0$. This condition ensures that $x(\alpha)$ always contains a contribution from *some* $x_k$, and hence its density $\rho(\alpha, \cdot)$ does not collapse to a Dirac measure at zero for any $\alpha$. In the following, we classify $\rho(\alpha, \cdot)$ as the solution to a set of continuity equations.

**Theorem 1** (Continuity equations). *For all $\alpha \in \Delta^K$, the probability distribution of the barycentric stochastic interpolant $x(\alpha)$ has a density $\rho(\alpha, x)$ which satisfies the $K + 1$ equations*

$$\partial_{\alpha_k} \rho(\alpha, x) + \nabla_x \cdot \big(g_k(\alpha, x)\rho(\alpha, x)\big) = 0, \qquad k = 0, \ldots, K. \tag{7}$$

*Above, each $g_k(\alpha, x)$ is defined as the conditional expectation*

$$g_k(\alpha, x) = \mathbb{E}[x_k | x(\alpha) = x], \qquad k = 0, \ldots, K, \tag{8}$$

*where $\mathbb{E}[x_k | x(\alpha) = x]$ denotes an expectation over $\rho_0(x_0)\rho(x_1, \ldots, x_k)$ conditioned on the event $x(\alpha) = x$. The score along each two-marginal path connected to $\rho_0$ is given by*

$$\forall \alpha_0 \neq 0 : \quad \nabla_x \log \rho(\alpha, x) = -\alpha_0^{-1} g_0(\alpha, x). \tag{9}$$

*Moreover, each $g_k$ is the unique minimizer of the objective*

$$L_k(\hat{g}_k) = \int_{\Delta^K} \mathbb{E}\big[|\hat{g}_k(\alpha, x(\alpha))|^2 - 2x_k \cdot \hat{g}_k(\alpha, x(\alpha))\big] d\alpha, \quad k = 0, \ldots, K, \tag{10}$$

*where the expectation $\mathbb{E}$ is taken over $(x_0, \ldots, x_K) \sim \rho_0(x_0)\rho(x_1, \ldots, x_K)$.*

The proof of Theorem 1 is given in Appendix A: it relies on the definition of the density $\rho(\alpha, x)$ that says that, for every suitable test function $\phi(x)$, we have

$$\int_{\mathbb{R}^d} \phi(x)\rho(\alpha, x)dx = \int_{\mathbb{R}^{(K+1)d}} \phi(x(\alpha))\rho_0(x_0)\rho(x_1, \ldots, x_K)dx_0 \cdots dx_K. \tag{11}$$

where $x(\alpha) = \sum_{k=0}^{K} \alpha_k x_k$. Taking the derivative of both sides with respect to $\alpha_k$ using the chain rule as well as $\partial_{\alpha_k} x(\alpha) = x_k$, gives us

$$\int_{\mathbb{R}^d} \phi(x)\partial_{\alpha_k} \rho(\alpha, x)dx = \int_{\mathbb{R}^{(K+1)d}} x_k \cdot \nabla\phi(x(\alpha))\rho_0(x_0)\rho(x_1, \ldots, x_K)dx_0 \cdots dx_K. \tag{12}$$

Integrating the right hand side by parts and using the definition of the conditional expectation implies (7).

As we will see, the loss functions in (10) are amenable to empirical estimation over a dataset of samples, which enables efficient learning of the $g_k$. The resulting approximations can be combined according to (14) to construct a multimarginal generative model. In practice, we have the option to parameterize a single, weight-shared $g(\alpha, x) = (g_0(\alpha, x), \ldots g_K(\alpha, x))$ as a function from $\Delta^{(K-1)} \times \mathbb{R}^d \to \mathbb{R}^{K \times d}$, or to parameterize the $g_k : \Delta^{(K-1)} \times \mathbb{R}^d \to \mathbb{R}^d$ individually. In our numerical experiments, we proceed with the former for efficiency.

Theorem 1 provides relations between derivatives of the density $\rho(\alpha, x)$ with respect to $\alpha$ and with respect to $x$ that involve the conditional expectations $g_k(\alpha, x)$. By specifying a curve $\alpha(t)$ that traverses the simplex, we can use this result to design generative models that transport directly from any one marginal to another, or that are influenced by multiple marginals throughout the generation process, as we now show.

**Corollary 2** (Transport equations). *Let $\{e_k\}$ represent the standard basis vectors of $\mathbb{R}^{K+1}$, and let $\alpha : [0,1] \to \Delta^K$ denote a differentiable curve satisfying $\alpha(0) = e_i$ and $\alpha(1) = e_j$ for any $i, j = 0, \ldots, K$. Then the barycentric stochastic interpolant $x(\alpha(t))$ has probability density $\bar{\rho}(t, x) = \rho(\alpha(t), x)$ that satisfies the transport equation*

$$\partial_t \bar{\rho}(t, x) + \nabla \cdot \big(b(t, x)\bar{\rho}(t, x)\big) = 0, \qquad \bar{\rho}(t = 0, x) = \rho_i(x), \qquad \bar{\rho}(t = 1, x) = \rho_j(x), \tag{13}$$

*where we have defined the velocity field*

$$b(t, x) = \sum_{k=0}^{K} \dot{\alpha}_k(t) g_k(\alpha(t), x). \tag{14}$$

*In addition, the probability flow associated with* (13) *given by*

$$\dot{X}_t = b(t, X_t) \tag{15}$$

*satisfies* $X_{t=1} \sim \rho_j$ *for any* $X_{t=0} \sim \rho_i$*, and vice-versa.*

Corollary 2 is proved in Appendix A, where we also show how to use the information about the score in (9) to derive generative models based on stochastic dynamics. The transport equation (13) is a simple consequence of the identity $\partial_t \bar{\rho}(t, x) = \sum_{k=0}^{K} \dot{\alpha}_k(t) \partial_{\alpha_k} \rho(\alpha(t), x)$, in tandem with the equations in (7). It states that the barycentric interpolant framework leads to a generative model defined for any path on the simplex, which can be used to transport between any pair of marginal densities.

Note that the two-marginal transports along fixed edges in (13) reduce to a set of $K$ independent processes, because each $g_k$ takes values independent from the rest of the simplex when restricted to an edge. In fact, for any edge between $\rho_i$ and $\rho_j$ with $i, j \neq k$, the marginal vector field $g_k(\alpha, x) = \mathbb{E}[x_k | x(\alpha) = x] = \mathbb{E}[x_k]$, because the conditioning event provides no additional information about $x_k$. In practice, however, we expect imperfect learning and the implicit bias of neural networks to lead to models with two-marginal transports that are influenced by all data used during training. We summarize a few possibilities with the multimarginal generative modeling setup in Section 2.1.

## 2.1 OPTIMIZING THE PATHS TO LOWER THE TRANSPORT COST

Importantly, the results above highlight that the learning problem for the velocity $b$ is decoupled from the choice of path $\alpha(t)$ on the simplex. We can consider the extent to which this flexibility can lower the cost of transporting a given $\rho_i$ to another $\rho_j$ when this transport is accomplished by solving (15) subject to (13). In particular, we are free to parameterize the $\alpha_k$ in the expression for the velocity $b(t, x)$ given in (14) so long as the simplex constraints on $\alpha$ are satisfied. We use this to state the following corollary.

**Corollary 3.** *The solution to*

$$\mathcal{C}(\hat{\alpha}) = \min_{\hat{\alpha}} \int_0^1 \mathbb{E}\Big[\Big| \sum_{k=0}^{K} \dot{\hat{\alpha}}_k(t) g_k(\hat{\alpha}(t), x(\hat{\alpha}(t))) \Big|^2 \Big] dt \tag{16}$$

*gives the transport with least path length in Wasserstein-2 metric over the class of velocities* $\hat{b}(t, x) = \sum_{k=0}^{K} \dot{\hat{\alpha}}_k(t) g_k(\hat{\alpha}(t), x)$*. Here* $g_k$ *is given by* (8)*,* $x(\alpha) = \sum_{k=0}^{K} \alpha_k x_k$*, the expectation is taken over* $(x_0, x_1 \ldots x_K) \sim \rho_0 \cdot \rho$*, and the minimization is over all paths* $\hat{\alpha} \in C^1([0, 1])$ *such that* $\hat{\alpha}(t) \in \Delta^K$ *for all* $t \in [0, 1]$*.*

Note the objective in (16) can be estimated efficiently without having to simulate the probability flow given by (15) by sampling the interpolant. Note also that this gives a way to reduce the transport directly without having to solve the max-min problem presented (Albergo & Vanden-Eijnden, 2023). We stress however that this class of velocities is not sufficient to achieve optimal transport, as such a guarantee requires the use of nonlinear interpolants (cf. Appendix D of Albergo & Vanden-Eijnden (2023)). An example learnable parameterization is given in Appendix C.1.

In addition to optimization over the path, the transport equation (13) enables generative procedures that are unique to the multimarginal setting, such as transporting any given density to the barycenter of the $K + 1$ densities, or transport through the interior of the simplex.

## 3 NUMERICAL EXPERIMENTS

In this section, we numerically explore the characteristics, behavior, and applications of the multimarginal generative model introduced above.

---

**Algorithm 1:** Learning each $\hat{g}_k$

**Input:** model $\hat{g}_k$, coupling $\rho(x_1, \ldots, x_k)$, gradient steps $N_g$, loss function $L_k$ as defined in (10)

**for** $j = 1, \ldots, N_g$ **do**
  draw $(x_0, \cdots x_K) \sim \rho_0(x_0)\rho(x_1, \cdots, x_K)$
  draw $\alpha \sim \Delta^K$
  make $x(\alpha) = \sum_k \alpha_k x_k$
  Take gradient step with respect to $L_k$
**end**

**Return:** $\hat{g}_k$.

---

| Example | Example Application | | Simplex representation |
|---|---|---|---|
| 2-marginal | standard generative modeling | $\rho_1 \to \rho_2$ | |
| 3-marginal | two-marginal with smoothing | $\rho_1 \to \rho_2$ | |
| K-marginal | all-to-all image translation/style transfer | $\rho_i \to \rho_j$ | |

Table 1: Characterizing various generative modeling aims with respect to how their transport appears on the simplex. We highlight a sampling of various other tasks that can emerge from generic multimarginal generative modeling. Here, two-marginal with smoothing corresponds to the addition of a Gaussian marginal, as done in (Albergo et al., 2023), with a path through the simplex.

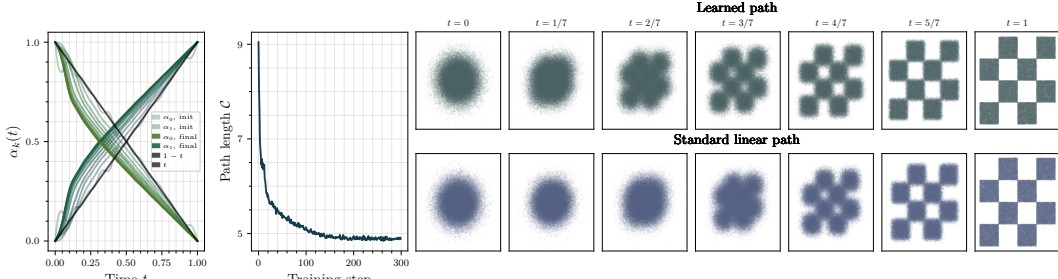

Figure 1: Direct optimization of $\alpha(t)$ over a parametric class to reduce transport cost in the 2-marginal learning problem of a Gaussian to the checkerboard density. *Left*: The initial and final $\hat{\alpha}_0, \hat{\alpha}_1$ learned over 300 optimizations steps on (16). *Center*: The reduction in the path length over training. *Right*: Time slices of the probability density $\bar{\rho}(t, x)$ corresponding to learned interpolant with learned $\hat{\alpha}$ as compared to the linear interpolant $\alpha = [1 - t, t]$.

We start with the numerical demonstration of the path minimization in the restricted class defined in Corollary 3. Following that, we explore various pathways of transport on the simplex to illustrate various characteristics and applications of the method. We explore what happens empirically with the respect to transport on the edges of the simplex and from its barycenter. We also consider empirically if probability flows following (15) with different paths specified by $\alpha(t)$ but the same endpoint $\alpha(1)$ on give meaningfully similar but varying samples. Algorithms for learning the vector fields $g_k$ and learning a least-cost parametric path (if desired) on the simplex are given in Algorithm 1 and Algorithm 2 respectively.

### 3.1 Optimizing the simplex path $\alpha(t)$

As shown in Section 2, the multimarginal formulation reveals that transport between distributions can be defined in a path-independent manner. Moreover, Corollary 3 highlights that the path can be determined *a-posteriori* via an optimization procedure. Here we study to what extent this optimization can lower the transport cost, and in addition, we consider what effect it has on the resulting path at the qualitative level of the samples. We note that while we focus here on the stochastic interpolant framework considered in (5), score-based diffusion (Song et al., 2021c) fits within the original interpolant formalism as presented in (1) . This means that the frame-

---

**Algorithm 2:** Learning a path $\hat{\alpha(t)}$

**Input:** model $\hat{g}_k$, model path $\hat{\alpha}(t)$, coupling $\rho(x_1, \ldots, x_k)$, gradient steps $N_{\text{steps}}$, loss function $\mathcal{C}(\hat{\alpha})$ as defined in (16)

**for** $j = 1, \ldots, N_g$ **do**
   draw $(x_0, \cdots x_K) \sim \rho_0(x_0)\rho(x_1, \cdots, x_K)$
   draw $t \sim \text{Unif}[0, 1]$
   compute $\hat{\alpha}(t)$
   make $x(\alpha(t)) = \sum_k \alpha_k(t)x_k$
   Take gradient step with respect to $\mathcal{C}(\hat{\alpha})$
**end**
**Return:** $\hat{\alpha}(t)$.

work that we consider can also be used to optimize over the diffusion coefficient $g(t)$ for diffusion models, providing an algorithmic procedure for the experiments proposed by Karras et al. (2022).

A numerical realization of the effect of path optimization is provided in Fig. 1. We train an interpolant on the two-marginal problem of mapping a standard Gaussian to the two-dimensional checkerboard density seen in the right half of the figure. We parameterize the $\alpha_0$ and $\alpha_1$ in this case with a Fourier expansion with $m = 20$ components normalized to the simplex; the complete formula is given in Appendix C.1. A visualization of the improvements from the learned $\alpha$ for fixed number of function evaluations is also given in the appendix. We note that this procedure yields $\alpha_0$ and $\alpha_1$ that are qualitatively very similar to those observed by Shaul et al. (2023), who study how to minimize a kinetic cost of Gaussian paths to the checkerboard.

## 3.2 ALL-TO-ALL IMAGE TRANSLATION AND STYLE TRANSFER

A natural application of the multimarginal framework is image-to-image translation. Given $K$ image distributions, every image dataset is mapped to every other in a way that can be exploited, for example, for style transfer. If the distributions have some distinct characteristics, the aim is to find a map that connects samples $x_i$ and $x_j$ from $\rho_i$ and $\rho_j$ in a way that visually transforms aspects of one into another. We explore this in two cases. The first is a class to class example on the MNIST dataset, where every digit class is mapped to every other digit class. The second is a case using multiple common datasets as marginals: the AFHQ-2 animal faces dataset (Choi et al., 2020), the Oxford flowers dataset (Nilsback & Zisserman, 2006), and the CelebA dataset (Zhang et al., 2019). All three are set to resolution $64 \times 64$. For each case, we provide pictorial representations that indicate which marginals map where using either simplex diagrams or Petrie polygons representing the edges of the higher-order simplices. For all image experiments, we use the U-Net architecture made popular in (Ho et al., 2020).

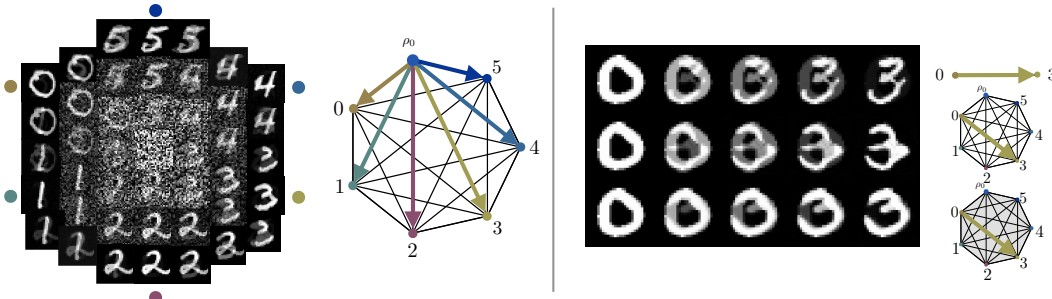

Figure 2: Left: Generated MNIST digits from the same Gaussian sample $x_0 \sim \rho_0$, with $K = 7$ marginals ($\rho_0$ and 6 digit classes). $x_0$ is visualized in the center of the image collection at time $t = 0$, and the perimeter corresponds to transport to the edge of the simplex at time $t = 1$ with vertices color-coded. A Petrie polygon representing the 6-simplex, with arrows denoting transport from the Gaussians along edges to the color-coded marginals, clarifies the marginal endpoints. Right: Demonstrating the impact of learning with over the larger simplex. Top row: learning just on the simplex edge from 0 to 3. Middle: Learning on all the simplex edges from 0 through 5. Bottom: Learning on the entire simplex constructed from 0 through 5 and not just the edges.

**Qualitative ramifications of multimarginal transport**  We use mappings between MNIST digits as a simple testbed to begin a study of multimarginal mappings. For visual simplicity, we consider the case of a set of marginals containing the digits 0 through 5 and a Gaussian. The inclusion of the Gaussian marginal facilitates resampling to generate from any given marginal, in addition to the ability to map from any $\rho_i$ to any other $\rho_j$.

In the left of Fig. 2, we show the generation space across the simplex for samples pushed forward from the same Gaussian sample towards all other vertices of the simplex. Directions from the center to the colored circles indicate transport towards one of the vertices corresponding to the different digits. We note that this mapping is smooth amongst all the generated endpoint digits, allowing for a natural translation toward every marginal.

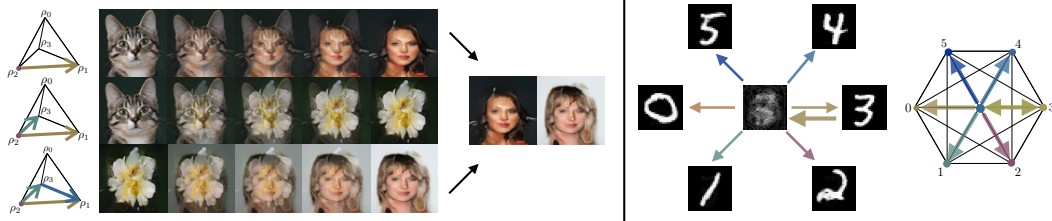

Figure 3: Left: An illustration of how different transport paths on the simplex can reach final samples that have similar content. Top row: A cat sample is transformed into a celebrity. Middle row: The same cat is pushed to the flower marginal. Bottom row: The new marginal flower sample is then pushed to a celebrity that maintains meaningful semantic structure from the celebrity generated from along the other path on the simplex.

On the right of this figure, we illustrate how training on the simplex influences the learned generative model. The top row corresponds to learning just in the two-marginal case. The middle row corresponds to learning only on the *edges* of the simplex, and is equivalent to learning multiple two-marginal models in one shared representation. The third corresponds to learning on the entire simplex. We observe here (and in additional examples provided in Appendix C.2) that learning on the whole simplex empirically results in image translation that better preserves the characteristics of the original image. This can best be seen in the bottom row, where the three naturally emerges from the shape of the zero.

In addition to translating *along* the marginals, one can try to forge a correspondence between the marginals by pushing samples through the barycenter of the simplex defined by $\alpha(t) = (\frac{1}{K+1}, \ldots, \frac{1}{K+1})$. This is a unique feature of the multimarginal generative model, and we demonstrate its use in the right of Fig. 3 to construct other digits with a correspondence to a given sample 3.

We next turn to the higher-dimensional image generation tasks depicted in Fig. 4, where we train a 4-marginal model ($K = 3$) to generate and translate among the celebrity, flowers, and animal face datasets mentioned above. In the left of this figure we present results generated marginally from the Gaussian vertex for a model trained on the entire simplex. Depictions of the transport along edges of the simplex are shown to the right. We show in the right half of the figure that image-to-image translation between the marginals has the capability to pick up coherent form, texture, and color from a sample from a sample from one marginal and bring it to another (e.g. the wolf maps to a flower with features of the wolf). Additional examples are available in Appendix C.2.

As a final exploration, we consider the effect of taking different paths on the simplex to the same vertex. As mentioned in the related works and in Theorem 4, a coupling of the form (31) would

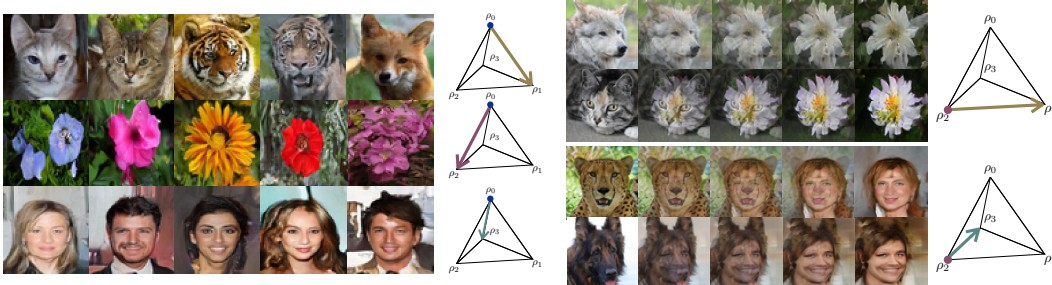

Figure 4: Left: Marginally sampling a $K = 4$ multimarginal model comprised of the AFHQ, flowers, and CelebA datasets at resolution $64 \times 64$. Shown to the right of the images is the corresponding path taken on the simplex, with $\alpha(0) = e_0$ starting at the Gaussian $\rho_0$ and ending at one of $\rho_1, \rho_2$ or $\rho_3$. Right: Demonstration of style transfer that emerges naturally when learning a multimarginal interpolant. With a single shared interpolant, we flow from the AFHQ vertex $\rho_2$ to the flowers vertex $\rho_1$ or to the CelebA vertex $\rho_3$. The learned flow connects images with stylistic similarities.

lead to sample generation independent of the interior of the path $\alpha(t)$ and dependent only on its endpoints. Here, we do not enforce such a constraint, and instead examine the path-dependence of an unconstrained multimarginal model. In the left of Fig. 3, we take a sample of a cat from the animals dataset and push it to a generated celebrity. In the middle row, we take the same cat image and push it toward the flower marginal. We then take the generated flower and push it towards the celebrity marginal. We note that there are similarities in the semantic structure of the two generated images, despite the different paths taken along the simplex. Nevertheless, the loop does not close, and there are differences in the generated samples, such as in their hair. This suggests that different paths through the space of measures defined on the simplex have the potential to provide meaningful variations in output samples using the same learned marginal vector fields $g_k$.

## 4    OUTLOOK AND CONCLUSION

In this work, we introduced a generic framework for multimarginal generative modeling. To do so, we used the formalism of stochastic interpolants, which enabled us to define a stochastic process on the simplex that interpolates between any pair of $K$ densities at the level of the samples. We showed that this formulation decouples the problem of learning a velocity field that accomplishes a given dynamical transport from the problem of designing a path between two densities, which leads to a simple minimization problem for the path with lowest transport cost over a specific class of velocity fields. We considered this minimization problem in the two-marginal case numerically, and showed that it can lower the transport cost in practice. It would be interesting to apply the method to score-based diffusion, where significant effort has gone into heuristic tuning of the signal to noise ratio (Karras et al., 2022), and see if the optimization can recover or improve upon recommended schedules. In addition, we explored how training multimarginal generative models impacts the generated samples in comparison to two-marginal couplings, and found that the incorporation of data from multiple densities leads to novel flows that demonstrate emergent properties such as style transfer. Future work will consider the application of this method to problems in measurement decorruption from multiple sources.

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

## A   Omitted proofs

**Theorem 1** (Continuity equations)**.** *For all $\alpha \in \Delta^K$, the probability distribution of the barycentric stochastic interpolant $x(\alpha)$ has a density $\rho(\alpha, x)$ which satisfies the $K + 1$ equations*

$$\partial_{\alpha_k} \rho(\alpha, x) + \nabla_x \cdot \big( g_k(\alpha, x) \rho(\alpha, x) \big) = 0, \qquad k = 0, \ldots, K. \tag{7}$$

*Above, each $g_k(\alpha, x)$ is defined as the conditional expectation*

$$g_k(\alpha, x) = \mathbb{E}[x_k | x(\alpha) = x], \qquad k = 0, \ldots, K, \tag{8}$$

*where $\mathbb{E}[x_k | x(\alpha) = x]$ denotes an expectation over $\rho_0(x_0)\rho(x_1, \ldots, x_k)$ conditioned on the event $x(\alpha) = x$. The score along each two-marginal path connected to $\rho_0$ is given by*

$$\forall \alpha_0 \neq 0 : \quad \nabla_x \log \rho(\alpha, x) = -\alpha_0^{-1} g_0(\alpha, x). \tag{9}$$

*Moreover, each $g_k$ is the unique minimizer of the objective*

$$L_k(\hat{g}_k) = \int_{\Delta^K} \mathbb{E}\big[ |\hat{g}_k(\alpha, x(\alpha))|^2 - 2 x_k \cdot \hat{g}_k(\alpha, x(\alpha)) \big] d\alpha, \quad k = 0, \ldots, K, \tag{10}$$

*where the expectation $\mathbb{E}$ is taken over $(x_0, \ldots, x_K) \sim \rho_0(x_0)\rho(x_1, \ldots, x_K)$.*

*Proof.* By definition of the barycentric interpolant $x(\alpha) = \sum_{k=0}^{K} \alpha_k x_k$, its characteristic function is given by

$$\mathbb{E}[e^{ik \cdot x(\alpha)}] = \int_{\mathbb{R}^d \times \mathbb{R}^d} e^{ik \cdot \sum_{k=1}^{K} x_k} \rho(x_1, \ldots, x_K) dx_1 \cdots dx_K e^{-\frac{1}{2}\alpha_0^2 |k|^2} \tag{17}$$

where we used $x_0 \perp (x_1, \ldots, x_K)$ and $x_0 \sim \mathsf{N}(0, Id_d)$. The smoothness in $k$ of this expression guarantees that the distribution of $x(\alpha)$ has a density $\rho(\alpha, x) > 0$. Using the definition of $x(\alpha)$ again, $\rho(\alpha, x)$ satisfies, for any suitable test function $\phi : \mathbb{R}^d \to \mathbb{R}$:

$$\int_{\mathbb{R}^d} \phi(x)\rho(t, x|\xi) dx = \int_{\mathbb{R}^d \times \cdots \times \mathbb{R}^d} \phi(x(\alpha)) \rho(x_1, \ldots, x_K)(2\pi)^{-d/2} e^{-\frac{1}{2}|x_0|^2} dx_0 \cdots dx_K. \tag{18}$$

Taking the derivative with respect to $\alpha_k$ of both sides we get

$$
\begin{aligned}
&\int_{\mathbb{R}^d} \phi(x) \partial_{\alpha_k} \rho(\alpha, x) dx \\
&= \int_{\mathbb{R}^d \times \cdots \times \mathbb{R}^d} x_k \cdot \nabla \phi(x(\alpha)) \rho(x_1, \ldots, x_K)(2\pi)^{-d/2} e^{-\frac{1}{2}|x_0|^2} dx_0 \cdots dx_K \\
&= \int_{\mathbb{R}^d} \mathbb{E}\big[ x_k \cdot \nabla \phi(x(\alpha)) \big| x(\alpha) = x \big] \rho(\alpha, x) dx \\
&= \int_{\mathbb{R}^d} \mathbb{E}[x_k | x(\alpha) = x] \cdot \nabla \phi(x) \rho(\alpha, x) dx
\end{aligned}
\tag{19}
$$

where we used the chain rule to get the first equality, the definition of the conditional expectation to get the second, and the fact that $\phi(x(\alpha)) = \phi(x)$ since we condition on $x(\alpha) = x$ to get the third. Since

$$\mathbb{E}[x_k | x(\alpha) = x] = g_k(\alpha, x) \tag{20}$$

by the definition of $g_k$ in (8), we can write (19) as

$$\int_{\mathbb{R}^d} \phi(x) \partial_{\alpha_k} \rho(\alpha, x) dx = \int_{\mathbb{R}^d} g_k(\alpha, x) \cdot \nabla \phi(x) \rho(\alpha, x) dx \tag{21}$$

This equation is (7) written in weak form.

To establish (9), note that, if $\alpha_0 > 0$, we have

$$\mathbb{E}\big[ x_0 e^{i\alpha_0 k \cdot x_0} \big] = -\alpha_0^{-1}(i\partial_k)\mathbb{E}\big[ e^{i\alpha_0 k \cdot x_0} \big] = -\alpha_0^{-1}(i\partial_k) e^{-\frac{1}{2}\alpha_0^2 |k|^2} = i\alpha_0 k e^{-\frac{1}{2}\alpha_0^2 |k|^2}. \tag{22}$$

As a result, using $x_0 \perp (x_1, \ldots, x_K)$, we have

$$\mathbb{E}\big[ x_0 e^{ik \cdot x(\alpha)} \big] = i\alpha_0 k \mathbb{E}\big[ e^{ik \cdot x(\alpha)} \big] \tag{23}$$

Using the properties of the conditional expectation, the left-hand side of this equation can be written as

$$
\begin{aligned}
\mathbb{E}\left[x_0 e^{ik \cdot x(\alpha)}\right] &= \int_{\mathbb{R}^d} \mathbb{E}\left[x_0 e^{ik \cdot x(\alpha)} \big| x(\alpha) = x\right] \rho(\alpha, x) dx \\
&= \int_{\mathbb{R}^d} \mathbb{E}[x_0 | x(\alpha) = x] e^{ik \cdot x} \rho(\alpha, x) dx \\
&= \int_{\mathbb{R}^d} g_0(\alpha, x) e^{ik \cdot x} \rho(\alpha, x) dx
\end{aligned}
\tag{24}
$$

where we used the definition of $g_0$ in (8) to get the last equality. Since the right-hand side of (23) is the Fourier transform of $-\alpha_0 \rho(\alpha, x)$, we deduce that

$$
g_0(\alpha, x) \rho(\alpha, x) = -\alpha_0 \nabla \rho(\alpha, x) = -\alpha_0 \nabla \log \rho(\alpha, x)\, \rho(\alpha, x).
\tag{25}
$$

Since $\rho(\alpha, x) > 0$, this implies (9) when $\alpha > 0$.

Finally, to derive (10), notice that we can write

$$
\begin{aligned}
L_k(\hat{g}_k) &= \int_{\Delta^K} \mathbb{E}\left[|\hat{g}_k(\alpha, x(\alpha))|^2 - 2x_k \cdot \hat{g}_k(\alpha, x(\alpha))\right] d\alpha, \\
&= \int_{\Delta^K} \int_{\mathbb{R}^d} \mathbb{E}\left[|\hat{g}_k(\alpha, x(\alpha))|^2 - 2x_k \cdot \hat{g}_k(\alpha, x(\alpha)) | x(\alpha) = x\right] \rho(\alpha, x) dx d\alpha \\
&= \int_{\Delta^K} \int_{\mathbb{R}^d} \left[|\hat{g}_k(\alpha, x(\alpha))|^2 - 2\mathbb{E}[x_k | x(\alpha)] \cdot g_k(\alpha, x(\alpha))\right] \rho((\alpha, x) dx d\alpha \\
&= \int_{\Delta^K} \int_{\mathbb{R}^d} \left[|\hat{g}_k(\alpha, x(\alpha))|^2 - 2g_k(t, x, \xi) \cdot \hat{g}_k(\alpha, x(\alpha))\right] \rho(\alpha, x) dx d\alpha
\end{aligned}
\tag{26}
$$

where we used the definition of $g_k$ in (8). The unique minimizer of this objective function is $\hat{g}_k(\alpha, x) = g_k(\alpha, x)$.

$\square$

**Corollary 2** (Transport equations). *Let $\{e_k\}$ represent the standard basis vectors of $\mathbb{R}^{K+1}$, and let $\alpha : [0,1] \to \Delta^K$ denote a differentiable curve satisfying $\alpha(0) = e_i$ and $\alpha(1) = e_j$ for any $i, j = 0, \ldots, K$. Then the barycentric stochastic interpolant $x(\alpha(t))$ has probability density $\bar{\rho}(t, x) = \rho(\alpha(t), x)$ that satisfies the transport equation*

$$
\partial_t \bar{\rho}(t, x) + \nabla \cdot \left(b(t, x) \bar{\rho}(t, x)\right) = 0, \qquad \bar{\rho}(t = 0, x) = \rho_i(x), \qquad \bar{\rho}(t = 1, x) = \rho_j(x), \tag{13}
$$

*where we have defined the velocity field*

$$
b(t, x) = \sum_{k=0}^{K} \dot{\alpha}_k(t) g_k(\alpha(t), x).
\tag{14}
$$

*In addition, the probability flow associated with (13) given by*

$$
\dot{X}_t = b(t, X_t)
\tag{15}
$$

*satisfies $X_{t=1} \sim \rho_j$ for any $X_{t=0} \sim \rho_i$, and vice-versa.*

*Proof.* By definition $\bar{\rho}(t = 0, x) = \rho(\alpha(0), x) = \rho(e_i, x) = \rho_i(x)$ and $\bar{\rho}(t = 1, x) = \rho(\alpha(1), x) = \rho(e_j, x) = \rho_j(x)$, so the boundary conditions in (13) are satisfied. To derive the transport equation in (13) use the chain rule as well as (7) to deduce

$$
\partial_t \bar{\rho}(t, x) = \sum_{k=0}^{K} \dot{\alpha}_k(t) \partial_{\alpha_k} \bar{\rho}(\alpha, x) = -\sum_{k=0}^{K} \dot{\alpha}_k(t) \nabla \cdot \left(g_k(\alpha(t), x) \bar{\rho}(\alpha, x)\right).
\tag{27}
$$

This gives (13) by definition of $b(t, x)$ in (14). (15) is the characteristic equation associated with (13) which implies the statement about the solution of this ODE.
$\square$

Note that, using the expression in (13) for $\nabla \log \rho(\alpha(t), x) = \nabla \log \bar{\rho}(t, x)$ as well as the identity $\Delta \bar{\rho}(t, x) = \nabla \cdot (\nabla \bar{\rho}(t, x) \bar{\rho}(t, x))$ we can, for any $\epsilon(t) \geq 0$, write (13) as the Fokker-Planck equation (FPE)

$$\partial_t \bar{\rho}(t, x) + \nabla \cdot \left( \left( b(t, x) - \epsilon(t) \alpha_0^{-1}(t) g_0(\alpha_0(t), x) \right) \bar{\rho}(\alpha, x) \right) = \epsilon(t) \Delta \bar{\rho}(t, x). \tag{28}$$

The SDE associated with this FPE reads

$$dX_t^F = \left( b(t, X_t^F) - \epsilon(t) \alpha_0^{-1}(t) g_0(\alpha_0(t), X_t^F) \right) dt + \sqrt{2\epsilon(t)} dW_t. \tag{29}$$

As a result of the property of the solution $\bar{\rho}(t, x)$ we deduce that the solutions to the SDE (29) are such that $X_{t=1}^F \sim \rho_j$ if $X_{t=0}^F \sim \rho_i$, which results in a generative model using a diffusion.

**Corollary 3.** *The solution to*

$$\mathcal{C}(\hat{\alpha}) = \min_{\hat{\alpha}} \int_0^1 \mathbb{E}\left[ \left| \sum_{k=0}^K \dot{\hat{\alpha}}_k(t) g_k(\hat{\alpha}(t), x(\hat{\alpha}(t))) \right|^2 \right] dt \tag{16}$$

*gives the transport with least path length in Wasserstein-2 metric over the class of velocities $\hat{b}(t, x) = \sum_{k=0}^K \dot{\hat{\alpha}}_k(t) g_k(\hat{\alpha}(t), x)$. Here $g_k$ is given by (8), $x(\alpha) = \sum_{k=0}^K \alpha_k x_k$, the expectation is taken over $(x_0, x_1 \ldots x_K) \sim \rho_0 \cdot \rho$, and the minimization is over all paths $\hat{\alpha} \in C^1([0, 1])$ such that $\hat{\alpha}(t) \in \Delta^K$ for all $t \in [0, 1]$.*

*Proof.* The statement follows from the fact that the integral at the right-hand side of (16) can be written as

$$\int_0^1 \mathbb{E}\left[ \left| \sum_{k=0}^K \dot{\hat{\alpha}}_k(t) g_k(\hat{\alpha}(t), x(\hat{\alpha}(t))) \right|^2 \right] dt = \int_0^1 \mathbb{E}\left[ \left| \hat{b}(t, x(\hat{\alpha}(t))) \right|^2 \right] dt$$
$$= \int_0^1 \int_{\mathbb{R}^d} |b(t, x)|^2 \hat{\rho}(t, x) dx dt \tag{30}$$

where $\hat{b}(t, x) = \sum_{k=0}^K \dot{\hat{\alpha}}_k(t) g_k(\hat{\alpha}(t), x)$ and $\hat{\rho}(t, x)$ solves the transport equation (13) with $b$ replaced by $\hat{b}$. This expression give the length in Wasserstein-2 metric of the path followed by this density, implying the statement of the corollary.

$\square$

# B DETERMINISTIC COUPLINGS AND MAP DISTILLATION

We now describe an illustrative coupling $\rho_0(x_0)\rho(x_1, \ldots, x_K)$ for the multimarginal setting in which the probability flow between marginals is given in one step. Let $T_k : \mathbb{R}^d \to \mathbb{R}^d$ for $k = 1, \ldots, K$ be invertible maps such that $x_k = T(x_0) \sim \rho_k$ if $x_0 \sim \rho_0$ (i.e., each $\rho_k$ is the pushforward of $\rho_0$ by $T_k$). Assume that the coupling $\rho$ is of Monge type, i.e.,

$$\rho(x_1, \ldots, x_K) = \prod_{k=1}^K \delta(x_k - T_k(x_0)) \tag{31}$$

Clearly, this distribution satisfies (6). For simplicity of notation let us denote by $T_0(x) = x$ the identity map. Let us also assume that, for all $\alpha \in \Delta^K$, the map $\sum_{k=0}^k \alpha_k T_k(x)$ is invertible, with inverse $R(\alpha, x)$:

$$\forall \alpha \in \Delta^K, x \in \mathbb{R}^d \quad : \quad R\left( \alpha, \sum_{k=0}^K \alpha_k T_k(x) \right) = \sum_{k=0}^k \alpha_k T_k(R(\alpha, x)) = x \tag{32}$$

In this case, the factors $g_k(\alpha, x)$ evaluated at $\alpha = e_0$ simply recover the maps $T_k$. Evaluated at $e_i$, they transport samples from $\rho_i$ to samples from $\rho_k$, as shown in the following results:

**Theorem 4.** *Assume that $\rho(x_1, \ldots, x_K)$ is given by (31) and that (32) holds. Then, under the same conditions as in Corollary 2, the factors $g_k(\alpha, x)$ defined in (8) are given by*

$$g_k(\alpha, x) = T_k(R(\alpha, x)) \tag{33}$$

*In particular, if $\alpha = e_i$, we have $g_k(e_i, x) = T_k(T_i^{-1}(x))$ so that if $x_i \sim \rho_i$ then $g_k(e_i, x_i) \sim \rho_k$.*

*Proof.* If $\rho(x_1, \ldots, x_K)$ is given by (31), we have

$$g_k(\alpha, x) = \mathbb{E}_0\Big[T_k(x_0)\Big| \sum_{k=0}^{K} \alpha_k T_k(x_0) = x\Big] = \mathbb{E}_0\Big[T_k(x_0)\Big| x_0 = R(\alpha, x)\Big] = T_k(R(\alpha, x)). \quad (34)$$

where $\mathbb{E}_0$ denotes expectation over $x_0 \sim \rho_0$. The relation $g_k(e_i, x) = T_k(T_i^{-1}(x))$ follows from (33) since $R(e_i, x) = T_i^{-1}(x)$ and the final statement from (31). $\qquad\square$

In addition, the solution to the probability flow ODE (15) with initial data $X_{t=0} = T_i(x_0)$ is given by

$$X_t = \sum_{k=0}^{K} \alpha_k(t) T_k(x_0) \quad (35)$$

so that $X_{t=1} = T_j(x_0)$ since $\alpha(1) = e_j$ by assumption. To check (35), notice that $b(t, x) = \sum_{k=0}^{K} \dot{\alpha}_k(t) T_k(R(\alpha(t), x))$. As a result, using (32),

$$b(t, X_t) = \sum_{k=0}^{K} \dot{\alpha}_k(t) T_k\Big(R\Big(\alpha(t), \sum_{k'=0}^{K} \alpha_{k'}(t) T_{k'}(x_0)\Big)\Big) = \sum_{k=0}^{K} \dot{\alpha}_k(t) T_k(x_0) = \dot{X}_t \quad (36)$$

so that (35) satisfies (15).

## C   EXPERIMENTAL DETAILS

### C.1   TRANSPORT REDUCTION EXPERIMENTS

Corollary 3 tells us that, because the problem of learning to transport between any two densities $\rho_i$, $\rho_j$ on the simplex can separated from a specific path on the simplex, some of the transport costs associated to the probability flow (15) can be reduced by choosing the optimal $\alpha(t)$. $\alpha(t)$ can be parameterized as a learnable function conditional on this parameterization maintaining the simplex constraint $\sum_k \alpha_k = 1$.

In the 2D Gaussian to checkboard example provided in the main text, we fulfilled this by taking an $\tilde{\alpha}_i, \tilde{\alpha}_j$ of the form:

$$\tilde{\alpha}_i = 1 - t + \Big(\sum_{n}^{N} a_{i,n} \sin(n\tfrac{\pi}{2}t)\Big)^2 \quad (37)$$

$$\tilde{\alpha}_j = t + \Big(\sum_{n}^{N} a_{j,n} \sin(n\tfrac{\pi}{2}t)\Big)^2 \quad (38)$$

where $a_{i,n}, a_{j,n}$ are learnable parameters. We also found the square on the sum, though necessary for enforcing $a_{i,j} \geq 0$, could be removed if N is not too large for faster optimization. For the case when the simplex contains more than two marginals ($k \neq i, j$), the additional $\alpha_k$ which do not specify an endpoint $i, j$ of the transport can be parameterized drop the first terms which enforce the boundary conditions so that

$$\tilde{\alpha}_{k \neq i,j} = \Big(\sum_{n}^{N} a_{k,n} \sin(n\tfrac{\pi}{2}t)\Big)^2 \quad (39)$$

$$(40)$$

The $\alpha_k \, \forall k$ are normalized such that

$$\alpha = \Big[\tfrac{\tilde{\alpha}_0}{\sum \tilde{\alpha}_k}, \tfrac{\tilde{\alpha}_1}{\sum \tilde{\alpha}_k}, \ldots, \tfrac{\tilde{\alpha}_K}{\sum \tilde{\alpha}_k}\Big]. \quad (41)$$

As such, their time derivatives, necessary for using the ODE (15) as a generative model, is given as, for any component $\alpha_k$

$$\dot{\alpha}_k = \frac{\dot{\tilde{\alpha}}_k\big(\sum_{m \neq k} \tilde{\alpha}_m\big) - \tilde{\alpha}_k\big(\sum_{m \neq k} \dot{\tilde{\alpha}}_m\big)}{\big(\sum_k \tilde{\alpha}_k\big)^2}. \quad (42)$$

Learned path     Standard linear path

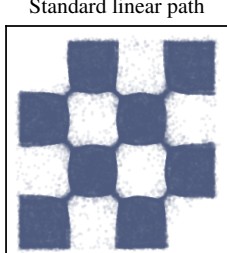

Figure 5: The output of the probability flow (15) realized by for the learned interpolant for the checkerboard problem discussed in Section 2.1 and Appendix C.1. For fixed number of function evaluations, 5 steps of the midpoint integrator, the learned $\alpha$ is quicker to give a more accurate solution.

## C.2 ARCHITECTURE FOR IMAGE EXPERIMENTS

We adapt the standard U-Net architecture (Ho et al., 2020) to work with a vector $\alpha$ describing the "time coordinate" rather than a scalar $t$, which is the conventional coordinate label. The number of output channels from the network is given as #image channels $\times K$, where each $k$th slice of #image channels corresponds to the $k$th marginal vector field necessary for doing the probability flow.

| | MNIST | AFHQ-Flowers-CelebA |
|---|---|---|
| Dimension | $32 \times 32$ | $64 \times 64 \times 3$ |
| # Training point | 50,000 | 15000, 8196, 200000 |
| Batch Size | 500 | 180 |
| Training Steps | $3 \times 10^5$ | $1 \times 10^5$ |
| Hidden dim | 128 | 256 |
| Attention Resolution | 64 | 64 |
| Learning Rate (LR) | 0.0002 | 0.0002 |
| LR decay (1k epochs) | 0.995 | 0.995 |
| U-Net dim mult | [1,2,2] | [1,2,3,4] |
| Learned $t$ sinusoidal embedding | Yes | Yes |
| $t_0, t_f$ when sampling with ODE | [0.0, 1.0] | [0.0, 1.0] |
| EMA decay rate | 0.9999 | 0.9999 |
| EMA start iteration | 10000 | 10000 |
| # GPUs | 2 | 8 |

Table 2: Hyperparameters and architecture for image datasets.

## C.3 ADDITIONAL IMAGE RESULTS

Here we provide some additional image results, applied to the image-to-image translation.

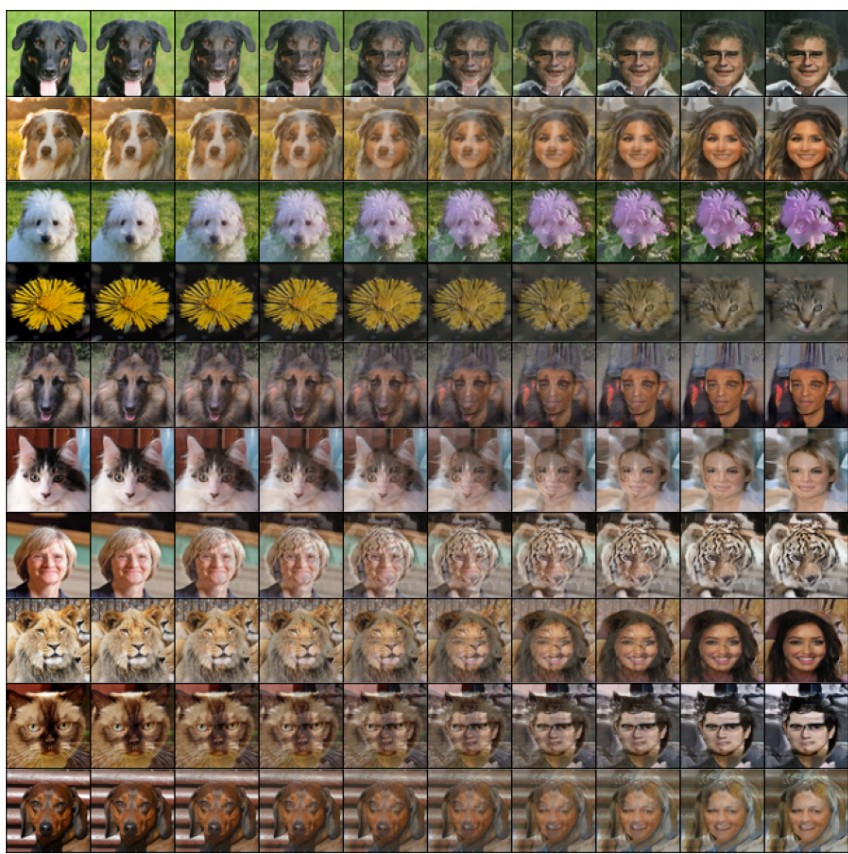

Figure 6: Random assortment of additional image translation examples built from the simplex of CelebA, the Oxford flowers dataset, and the animal faces dataset all at $64 \times 64$ resolution.

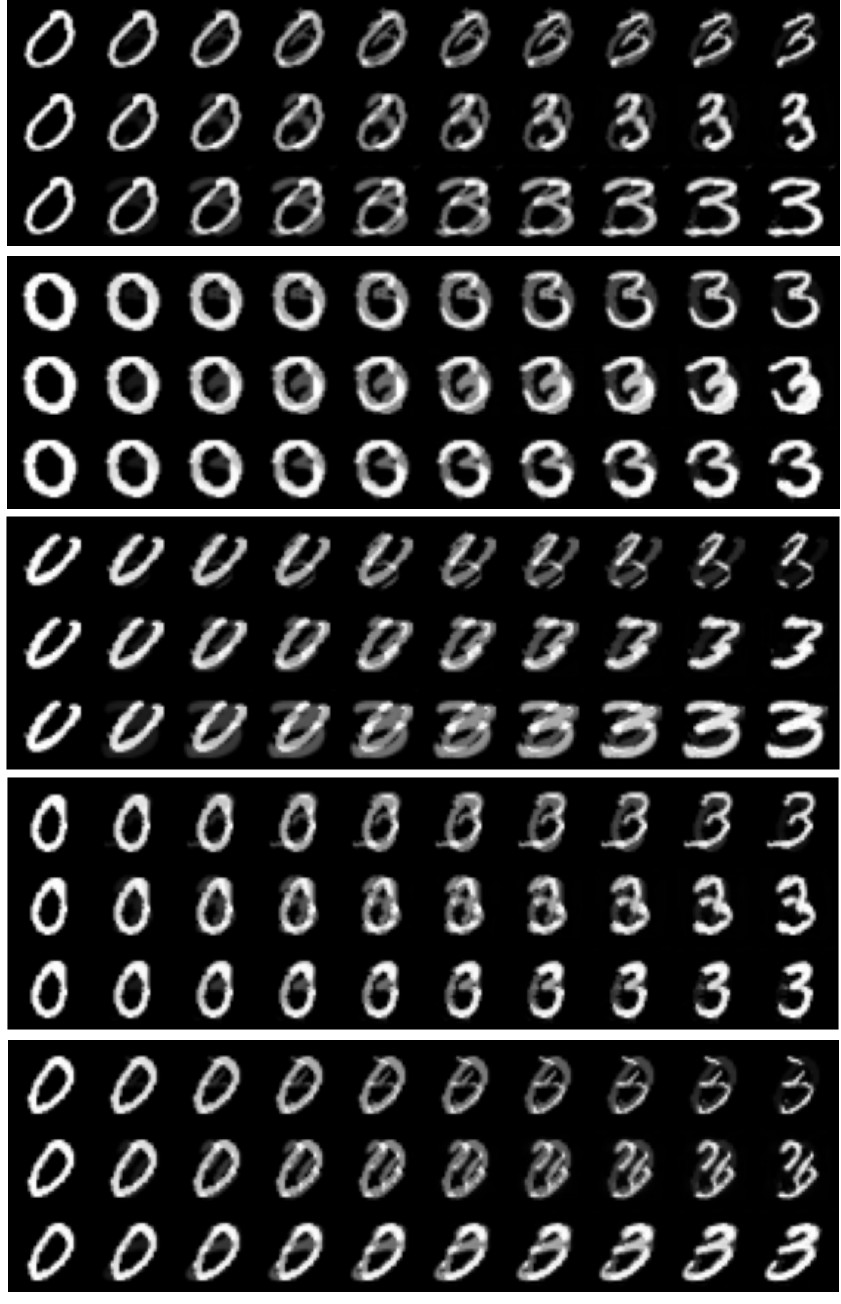

Figure 7: Random assortment of examples comparing the class-to-class translation of MNIST digits $0$ and $3$, where each row in a triplet corresponds to: top) model trained only on the edge between $0$ and $3$. middle) model trained on all edges between $0$ and $5$. bottom) model trained on the entire simplex.

