# OpenReview forum: "Multimarginal Generative Modeling with Stochastic Interpolants"
_ICLR.cc/2024/Conference — ICLR 2024 poster_

### Official Review · Reviewer_yfUs · 2023-10-13

**Soundness:** 3 good
**Presentation:** 3 good
**Contribution:** 2 fair
**Rating:** 5
**Confidence:** 3

**Summary:**

Multimarginal problems are typically studied in the context of optimal transport, by contrast, the authors studied the multimarginal problem using stochastic interpolant with a high-dimensional α in a simplex. The multimarginal framework allows us to (K, 2) marginal transport problems using only K marginal vector fields. Interesting experiments on all-to-all image translation are studied.

**Strengths:**

1. by lifting alpha to high-dimensional vector, the work enables to solve (K,2) marginal transport problems using only K marginal vector fields.

2. the overall presentation is clear and the empirical experiments are comprehensive.

**Weaknesses:**

Optimizing the vector field on simplex alpha may require delicate parameterizions. It would be better if it is more connected optimal transport.

The theories look like a na\"ive extension from the work of "Stochastic Interpolants: A Unifying Framework for Flows and Diffusions". The authors can kindly point out some innovations if I may miss some key points.

Section 2.2 may need some proper rewriting to be more clear.

**Questions:**

Table.1, what does the smoothing mean in "two-marginal with smoothing" ?


Minor:

typo? "color from a sample from a sample from one marginal"

---

> ### Author Response · Authors · 2023-11-21
> **Reply to reviewer yfUs**
>
> We thank reviewer yfUs for their feedback. We are glad that the reviewer found our presentation clear and our experiments comprehensive.  The reviewer, however, had concerns about the intrerpretability, utility, and novelty that we now address.
>
> ***Optimization on the simplex:*** The approach has precisely the advantage that it allows us to *solve the OT problem in a simulation free way on the simplex.* This is because we precompute (via quadratic regression) the conditional expectations $g_k(\alpha,x) = \mathbb{E}[x_k| x(\alpha)]$ where $x(\alpha) = \sum_{k=1}^K \alpha_k x_k$ and $x_k\sim\rho_k$. Having access to these conditional expectations allows us to calculate the transport cost along any transportation path parameterized by $\alpha(t)$ without having to perform any retraining and without having to solve an ODE: it is completely simulation-free in that respect. In turn, this allows us to optimize the path-parameterization $\alpha(t)$ to reduce the transportation cost. Of course, this only achieves OT within a specific parametric class  of velocities (i.e. those that transport samples on the simplex toward the PDF of any $x(\alpha) =  \sum_{k=1}^K \alpha_k x_k$), but it is still a nontrivial advance.
>
> ***Naïve extension:*** While the results follow quite naturally from the stochastic interpolant framework, they also constitute a nontrivial generalization of it. In particular, by showing that interpolation can be done in a multi-way fashion, we break from the standard perspective (used e.g. in score based diffusion models) in which the bridging betwen two distributions is itself defined via a time-dependent process (for example, a forward SDE that needs then to be reversed in time for generation). The original stochastic interpolant framework already showed that we can decouple the bridging of the densities from the generation process used afterwards; the multimarginal framework that we propose goes a step further in this direction by showing that the bridging itself does not need to use a  scalar time variable but rather can be made muti-way using a vector of $\alpha=(\alpha_1,\ldots,\alpha_K)$.
>
> ***Questions and typos***: Thanks for pointing out a few unclear phrases. In the text, we will clarify that "2-marginal with smoothing" refers to the case of having an interpolant $x(\alpha) = \alpha_0 x_0 + \alpha_1 x_1 + \alpha_2 z$ where $z$ is a Gaussian random variable that is independent of $(x_0, x_1)$. That is, there are three marginals, a density $\rho_0$, a density $\rho_1$, and a Gaussian density $\rho_2 = \rho_z$. For a parametric path in which $\alpha_2(t) > 0$ when $t \neq 0,1$, this corresponds to the Gaussian "smoothed" interpolant from [1]. On the simplex, it resembles the simplicial coordinate moving toward the top vertex of a triangle (which corresponds to $\rho_z$ in the illustration given in the table).

---

### Official Review · Reviewer_2Wkn · 2023-11-01

**Soundness:** 3 good
**Presentation:** 3 good
**Contribution:** 3 good
**Rating:** 5
**Confidence:** 4

**Summary:**

The paper introduces a vector-field-based generative model (incorporating ODE/SDE mechanisms) capable of generating samples from multiple marginal distributions. The methodology pivots around a generalized form of stochastic interpolant, specifically a barycentric stochastic interpolant. The stochastic process, in this context, is formulated as a weighted mix of samples from distinct marginal distributions. These weights, restricted to the simplex, allow the model to generate a weighted combination, defined as $x(\alpha)=\sum_{k=0}^K x_k \alpha_k$, where $\alpha=\left(\alpha_0, \ldots, \alpha_K\right) \in \Delta^K$. The corresponding probability flows are derived for this multi-weight interpolant path. To identify the optimal path with the lowest transport cost, the model is trained to minimize the Wasserstein-2 metric, considering the path's velocities. Consequently, this learned model allows transitions between various marginal distributions following the optimized path. However, the semantic interpretation of these paths, especially between distributions like images from multiple modes, remains ambiguous. In the experiments,

**Strengths:**

- *Innovative Generative Model*: The paper's proposition of a vector-field-based generative model that utilizes a generalized form of stochastic interpolant to generate from multiple marginals is unique and intriguing.
- *Optimal Path Identification*: By minimizing the Wasserstein-2 metric, the methodology seeks the most efficient path for transitions, with theoretical groundings.
- *Barycentric Stochastic Interpolant*: The use of a barycentric stochastic interpolant offers a nuanced approach, ensuring the weights adhere to a simplex and providing a structured method to learn to interpolate samples from different distributions.

**Weaknesses:**

- *Ambiguous Interpretation*: The paper does not provide a clear semantic understanding of the paths between distributions, making it difficult to ascertain the practical implications or the broader relevance of the methodology. Also, what does multi-marginal optimal transport path mean for image generation? Does it lead to better quality? In this paper, it is not answered.
- *Unknown Utility*: There's a lack of clarity on the potential use-cases for this generative model. For instance, if it's geared towards style transfer, it's essential to determine which style gets preserved and how it compares with existing baselines in that domain.
- *Novelty*: The novelty of doing stochastic interpolant for multiple marginal distributions is limited given that it is done for the two marginal case. The theoretical result seems to be an extension from the two marginal case to the multi-marginal case as well.
- *Minor*: missing caption for Figure 2 (right).

**Questions:**

- Can the authors provide some more justification on why learning such an optimal transport path is useful? If learning the path in between distributions is to facilitate objectives like style transfer, then it should be compared with baselines on that task and show the utility of having this model. Right now, I am struggling to understand what is the use case especially for images when there is not a well-defined joint distribution between the multiple marginals.
- It is unclear to me what is the implication of the Monge type coupling theoretical result. Is this shown to be relevant to the experimental results? If not I would recommend it to be put in appendix. It is also worth explaining more about the what this result implies and when we would like to have this property.

---

> ### Author Response · Authors · 2023-11-21
> **Reply tom reviewer 2Wkn**
>
> We thank reviewer 2Wkn for their feedback. We are glad that the reviewer found our multi-marginal stochastic interpolant framework unique and intriguing, and noted its connection with multi-marginal OT. The reviewer, however, had concerns about the interpretability, utility, and novelty, of the approach, which we now address.
>
> ***Interpretability:*** The approach rests on the calculation of the conditional expectations $g_k(\alpha,x) = \mathbb{E}[x_k| x(\alpha)=x]$ where $x(\alpha) = \sum_{k=1}^K \alpha_k x_k$ and $x_k\sim\rho_k$. These functions have a direct intepretation: they give the Bayes-optimal estimator of the sample $x_k$ given the information contained in the observation $x(\alpha)$. The multimarginal stochastic interpolant framework turns this static picture into a dynamical one by showing that we can generate samples from $\rho(\alpha)$, the PDF of $x(\alpha)$, at any value of $\alpha$, and transport them along any time-parameterized path $\alpha(t)$ using either an ODE or an SDE with tunable diffusion coefficient. In terms of style transfer, it shows, for example, how a sample from a given distribution (say of cats) can be continuously transformed into a sample from another distribution (say of flowers) in a way that is influenced by additional distributions (via a path through the interior of the simplex), thereby incorporating their "style" along the way. We give some examples of such style-transfering paths in the paper, though we agree that more research is needed to reveal the full potential of the approach. We stress that this notion of style transfer is inherently different than any two-way correspondence, as the learned transport is affected by all the marginals.
>
> ***Utility:*** The approach offers possibilities that cannot be achieved by current generative models that proceed between only two marginals. Because the method gives access to the vector fields $g_k(\alpha,x)$ for all values of $\alpha$, we can *a-posteriori* change the interpolation path by varying the functional form of $\alpha(t)$ and thereby produce many different generative models (all unbiased) without having to perform any retraining. In turn, this allows us to use these models for multiple tasks, or to optimize them according to several suitable criteria (OT being just one possible choice), as already discused in the first point about interpretability above.
>
> ***Novelty:*** While the results follow quite naturally from the stochastic interpolant framework, they also constitute a nontrivial generalization of it. In particular, by showing that interpolation can be done in a multi-way fashion, we break from the standard perspective (used e.g. in score based diffusion models) in which the bridging between two distributions is itself defined via a time-dependent process (for example a forward SDE that need then to be reversed in time for generation). The original stochastic interpolant framework already showed that we can decouple the bridging of the densities from the generation process used afterwards; the multimarginal framework that we propose goes a step further in this direction by showing that the bridging itself does not need to use a scalar time variable but rather can be made multi-way using a vector of $\alpha=(\alpha_1,\ldots,\alpha_K)$.
>
> **Question:**
>
> - While we agree that the meaning of a joint distribution between images of different type is somewhat unclear, this is a concept that is being explored in the two-marginal case and is at the core of any method that tries to compute the optimal transport path between these two distributions. This is useful, for instance, in terms of function evaluations along the transporting path since it makes this path straighter. Our approach allows for an algorithm that finds a better path which does not require simulating the transport, via equation (16), which is not available in the original interpolant presentation. Note also that this can be achieved *without any retraining* which is a desirable feature, and also applies in the standard two-marginal case.
>
> - We agree that the discussion about Monge coupling is somewhat tangential to our main point and will move it to an Appendix, as suggested by the reviewer. As a replacement, to improve the clarity, we are adding pseudocode for the main algorithms presented.
>
> Thanks again for your valuable insights.

---

### Official Review · Reviewer_PidH · 2023-11-07

**Soundness:** 4 excellent
**Presentation:** 3 good
**Contribution:** 4 excellent
**Rating:** 8
**Confidence:** 3

**Summary:**

In this work, the authors propose an extension of stochastic interpolants for multimarginal generative modeling, which involves learning a joint probability distribution that captures multiple marginal probability densities. The proposed framework is able to capture multi-way correspondences.

**Strengths:**

The work proposes a theoretically sound and practical approach based on stochastic interpolants for the multimarginal setting. The overall proposed scheme is mathematically sound and computationally more feasible than existing schemes. Extensive experiments are performed to illustrate the idea and different scenarios of the proposed method.

**Weaknesses:**

The overall algorithm of the proposed method might not be very clear or easy to follow, especially for mathematically less mature audience. I suggest the authors add the pseudocode of the main algorithm to improve clarity.

**Questions:**

Referring to the weaknesses part, how do you solve (16)?

Other remarks:
- Incomplete caption for Figure 2 (Right).
- Page 9 first paragraph: “from a sample from a sample”
- References: Song et al. (2021b, c) are repeated.

---

> ### Author Response · Authors · 2023-11-21
> **Reply to reviewer PidH**
>
> We thank reviewer PiDH for their feedback. We are glad that the reviewer found the work mathematically sound and more feasible than existing methods, and that they were happy with the experimentation.
>
> The reviewer asked us to clarify the algorithm, which we can happily do by:
>
> ***Providing pseudocode*** and making a clear distinction between the steps that are specific to: *(i) learning the vector fields* that make possible the multimarginal framework, and *(ii) using them as a generative model* to sample from some initial condition (such as one of the marginals) on the simplex and transport to some final condition (such as one of the other marginals, or, say, to the barycenter).
>
> ***Clarifying how we solve equation (16)***, which is the path minimization loss. This optimization is done separately from learning the vector fields $g_k(\alpha,x) = \mathbb{E} [x_k | x(\alpha)=x]$ with $x(\alpha) = \sum_{k=1}^K \alpha_k x_k$. Once we have learned the $g_k$,  we can *a-posteriori* change the interpolation path by varying the functional form of $\alpha(t)$. For example, if we want to transport from marginal $\rho_i$ to marginal $\rho_j$, we can take any  $\alpha(t)$ with initial condition $\alpha_k(0) = \delta_{k,i}$ and a final condition $\alpha_k(1) = \delta_{k,j}$. Any parameterization of $\alpha(t)$ that meets these boundary conditions is sufficient. In the Appendix we present one such parameterization via an expansion of Fourier coefficients, which has the advantage that it has very few parameters; alternatively we could also use a neural network to represent $\hat \alpha(t)$. *With a parameterization $\hat \alpha(t)$ and the learned $g_k$ in hand, we can evaluate equation (16) in a simulation free manner just by sampling the interpolant* $x(\alpha) \sim \rho(\alpha, x)$ *and performing gradient descent on the parameters of $\hat \alpha(t)$*. This optimization is extremely cheap and typically costs a minute fraction of the cost of learning the vector fields. We will make this clear with the addition of a pseudocode algorithm in the revision.

---

### Author Response · Authors · 2023-11-21
**General reply**

We thank the reviewers for their constructive feedback, which will help improve the clarity and exposition of our contribution. Below, we summarize the reviewers comments and supply a collective response to them. Please let us know if you are satisfied with our characterization of your reviews and if we have addressed your questions.

### Summary of Reviewer feedback

All reviewers stressed the soundness and comprehensiveness of our theoretical contributions. Reviewer 2Wkn found the approach to be an innovative and nuanced way of approaching the multimarginal problem. Reviewer PiDH also pointed out that our approach is computationally more feasible than existing methods, and, along with reviewer yfUS, found the experiments such as the all-to-all image translation extensive and interesting.

In our interpretation, the *primary criticism* from the reviewers is aimed at improving the clarity of our method. All reviewers pointed to some necessary adjustments to the exposition that could make the algorithms as well as the motivation of the problem more clear. In the following summary, we detail these clarifications, as well as how we are currently adjusting the text to incorporate this useful feedback.

### Summarizing our results and the differences with prior works:

The purpose of this paper is to provide an efficient learning algorithm for the multi-way correspondence between $K$ densities, and to do so from a *generative perspective*, rather than earlier consideration of this problem which considers data from these distributions statically.

In terms of ***theoretical novelty***: While the results follow quite naturally from the stochastic interpolant framework, they also constitute a nontrivial generalization of it. In particular, by showing that interpolation can be done in a multi-way fashion, we break from the standard perspective (used e.g. in score-based diffusion models) in which the bridging betwen two distributions is itself defined via a time-dependent process (for example a forward SDE that needs then to be reversed in time for generation). The original stochastic interpolant framework already showed that we can decouple the bridging of the densities from the generation process used afterwards; the multimarginal framework that we propse goes a step further in this direction by showing that the bridging itself does not need to use a  scalar time variable but rather can be made multi-way using a vector of $\alpha=(\alpha_1,\ldots,\alpha_K)$.

In terms of ***practical implementation***: Given samples $x_k\sim\rho_k$ from $K$ probability densities $\rho_k$, all that needs to be learned are the *conditional expectations* $g_k(\alpha,x)=\mathbb{E}[x_k | x(\alpha) = x]$ where $x(\alpha) = \sum_{k=1}^K \alpha_k x_k$. These conditional expectations can be learned by quadratic regression for all values of $\alpha = (\alpha_1,\ldots, \alpha_K)$ on the simplex using the loss given in equation (10) in the paper. Because the method gives access to the vector fields $g_k(\alpha,x)$ for all values of $\alpha$, we can *a-posteriori* change the interpolation path by varying the functional form of $\alpha(t)$ and thereby produce many different generative models (all unbiased) without having to perform any retraining. In turn, this allows us to use these models for multiple tasks, or to optimize them according to several suitable criteria (OT being just one possible choice).

In terms of ***utility***: The approach offers possibilities that cannot be achieved by current generative models (which consider only two marginals). Besides the examples already given in the paper, we provide here an additional application:

- Given samples from two densities $\rho_1$ and $\rho_2$, we can generate samples from a third density $\rho_3$ by blending the features of the first two using as initial data in the generative models $\alpha_1 x_1+ \alpha_2 x_2$ with $x_1 \sim\rho_1$ and $x_2 \sim \rho_2$. This can be done for any value of $\alpha_1, \alpha_2$ on the simplex, and for any path $\alpha(t)$ that reaches the vertex of $\rho_3$ at time $t=1$ (e.g. such that $\alpha(t=0) = (\alpha_1,\alpha_2,0)$ and $\alpha(t=1) = (0,0,1)$). In particular, it also allows us to optimize this path, for example according to the OT criterion. Further details on this optimization are given below.

### Adjustments to the exposition and motivation

To address the concern of the reviewer, we wil clarify the points above in the revised version of our manuscript. In addition, we will add pseudocode algorithms as requested by reviewer PidH.

We hope that, given these adjustments to account for your advice and suggestions, you will be willing to accordingly raise your scores. Thanks again.

---

### Author Response · Authors · 2023-11-23
**Revised paper posted, thanks again to reviewers**

Dear reviewers,

We have now uploaded the revised PDF that address your requested clarifications. We have added the two pseudocode algorithms — one for learning the vector fields that define the generative model and one for learning a least cost path $\alpha(t)$. We have also moved the Monge coupling discussion to the appendix, and polished the text to better elucidate our point regarding the multiway correspondence.
Please let us know if there is anything else we can do to address your feedback.

We hope that, given the updates and clarifications, you would be willing to update your scores.

Thank you again.

---

### Meta-Review · Area_Chair_sVeN · 2023-12-12

**Metareview:**

This paper generalizes the stochastic interplant (SI) approach to the multi-marginal settings. It offers two contributions: It learns $K$ neural vector fields and post-train a weighting curve $\alpha(t)$ to minimize transport cost.
The paper introduces a novel and potentially useful generalization of SI to the multi-marginal setting. The optimization of paths $\alpha$ is also a nice contribution. The experiments and practical evidence for usefulness of the suggested approach presented in the paper are somewhat underwhelming which made two of the reviewers slightly negative about this paper. However, one of the three reviewers provided strong support for the paper and their request for more clarity by adding a pseudo-code was addressed by the authors in the revision. Overall, it seems this paper passes the bar for publication as (what appears to be) a first work that generalize tractable flow method to the multimarginal setting.

**Justification For Why Not Higher Score:**

The main point here is, as also partially explained in the Meta Review, is that the practical usefulness of the method is not sufficiently demonstrated, and the novelty of the method, while exists and non-trivial, is not huge compared to the original SI and similar papers.

**Justification For Why Not Lower Score:**

This paper seems to be the first to provide a tractable algorithm for learning flow generative models with prescribed marginals. Although not providing rock solid evidence to the efficacy of the presented method, as a first step this seems as enough of a contribution to grant acceptance.

---

### Decision · Program_Chairs · 2024-01-16

Accept (poster)